# Integrated Temperature–Humidity Sensors for a Pouch-Type Battery Using 100% Printing Process

**DOI:** 10.3390/s24010104

**Published:** 2023-12-25

**Authors:** Gyeongseok Oh, Jae-Ho Sim, Mijin Won, Minhun Jung, Snigdha Paramita Mantry, Dong-Soo Kim

**Affiliations:** 1Department of Creative Convergence Engineering, Hanbat National University, Yuseong-ku, Daejeon 305-719, Republic of Korea; oks6125@naver.com (G.O.); poseidon96@naver.com (J.-H.S.); 0mj00@naver.com (M.W.); 2Research Institute of Printed Electronics & 3D Printing, Hanbat National University, Yuseng-ku, Daejeon 305-719, Republic of Korea; jungminhun@gmail.com (M.J.); snigdha.paramita012@gmail.com (S.P.M.)

**Keywords:** temperature sensors, humidity sensors, pouch battery, printing process, EV

## Abstract

The performance, stability, and lifespan of lithium-ion batteries are influenced by variations in the flow of lithium ions with temperature. In electric vehicles, coolants are generally used to maintain the optimal temperature of the battery, leading to an increasing demand for temperature and humidity sensors that can prevent leakage and short circuits. In this study, humidity and temperature sensors were fabricated on a pouch film of a pouch-type battery. IDE electrodes were screen-printed on the pouch film and humidity- and temperature-sensing materials were printed using a dispenser process. Changes in the capacitance of the printed Ag-CNF film were used for humidity sensing, while changes in the resistance of the printed PEDOT:PSS film were used for temperature sensing. The two sensors were integrated into a single electrode for performance evaluation. The integrated sensor exhibited a response of ΔR ≈ 0.14 to temperature variations from 20 °C to 100 °C with 20% RH humidity as a reference, and a response of ΔC ≈ 2.8 to relative humidity changes from 20% RH to 80% RH at 20 °C. The fabricated integrated sensor is expected to contribute to efficient temperature and humidity monitoring applications in various pouch-type lithium-ion batteries.

## 1. Introduction

Global warming is accelerating due to the continuous increase in CO_2_ emissions. Additionally, the development and demand for internal combustion engines are declining owing to pressures from the energy crisis, such as the depletion of fossil fuels and increased electricity consumption [1,2]. Electric vehicles (EVs) are being developed worldwide as the future mode of transportation. Moreover, the growing prevalence of smartphones, augmented reality, and smartwatches has led to an increasing demand for small, high-capacity batteries. Lithium-ion batteries (Li-ion) are the preferred battery type because of their high energy and current densities. However, these batteries are limited in terms of cost, stability, and aging, but research is being conducted to address these limitations [3,4]. The battery temperature directly influences the performance, stability, and lifespan of the battery packs [5,6,7]. At low temperatures, the flow of ions within the battery decreases, thereby reducing the output and capacity. Conversely, at high temperatures, the flow of ions increases, enhancing output and capacity. However, this increased ion flow can generate excessive heat, leading to risks such as explosions and fires. Therefore, it is a critical factor [8,9,10,11,12]. Most battery manufacturers incorporate coolant channels at the bottom of battery packs for cooling purposes. Liquid cooling is efficient but can result in short circuits owing to leaks from external impacts. Manufacturers typically use up to 1–2 thermistors per module for temperature monitoring or employ indirect methods such as voltage changes in cells or gas emission [13,14,15]. Additionally, methods such as module-level leak detection sensors and cooling tank-level monitoring are used for leak detection [16,17]. These temperature and humidity monitoring methods limit the ability to check the condition of individual cells, making immediate detection and prevention difficult. Humidity affects the performance of Li-ion batteries.

Several studies have been conducted to develop flexible temperature sensors, and there are primarily three types of temperature sensors: pyroelectric detectors, resistive temperature detectors (RTDs), and thermistors. Thermistors utilize the change in the resistance of the sensing material with temperature and are widely used because of their simple structure, excellent responsiveness, and broad operating range [18]. Poly(3,4-ethylenedioxythiophene):poly(styrenesulfonate) (PEDOT:PSS) is a promising candidate for fabricating wearable temperature sensors owing to its good chemical and thermal stability, outstanding mechanical properties, high electrochemical activity, and tunable electrical characteristics. The temperature-sensing mechanism of PEDOT:PSS involves an increase in charge carrier mobility with increasing temperature, leading to an increase in conductivity. This was confirmed through the measured resistance changes; however, the PSS in PEDOT:PSS reacted with moisture due to the presence of sulfonic acid groups (-SO_3_H) in its chemical structure. Song et al. developed and fabricated a highly sensitive flexible temperature sensor using PEDOT:PSS/PANI with a response time of 200 ms [19].

Cellulose nanofibers (CNF) are biodegradable and contain numerous hydroxyl groups on their molecular chains. The humidity-sensing mechanism of CNF leveraged the fact that water molecules easily interact via hydrogen bonding with the OH groups and, consequently, vary the electrical permittivity of the material. Rivadeneyra et al. developed and fabricated a biodegradable moisture sensor using a CNF film with a sensitivity of −0.10 Ω/%RH [20,21]. Peter et al. and Ziyang et al. investigated the performance change in Li-ion batteries based on the relative humidity (RH) [22,23]. Using humidity sensors instead of leak detection sensors not only aids leak detection but also enables more precise humidity monitoring. Existing monitoring systems have separate temperature and moisture sensors that make them bulky [24,25]. Therefore, integrated temperature and humidity sensors are required for simultaneous operation. Such integrated sensors do not require additional circuits or boards, thus reducing the space requirements [26]. Moreover, the utilization of artificial intelligence (AI) can expand research on battery pack life, performance prediction, and management by processing data from numerous sensors. Li-ion batteries can be precisely predicted using AI methods and can be combined with a battery management system to improve EV performance. Khawaja et al. have conducted a comparative analysis between linear, random forest, gradient boost, and light-gradient boosting machines and concluded that the random forest model can accurately estimate the Li-ion battery state [27].

In this study, we investigated an integrated temperature and humidity sensor designed on the surface of a pouch film using a full printing process. We utilized a pouch film from pouch-type batteries as the substrate, employing its formability and high energy density to fabricate an interdigitated electrode (IDE) through screen printing. PEDOT: PSS was used for the temperature sensing. To enhance humidity stability, crosslinkers were added to improve performance, and a polydimethylsiloxane (PDMS) passivation layer coating was applied to prevent moisture penetration. Polymer-based temperature-sensing materials have the drawback of low thermal conductivity, which results in slow sensor response and recovery times. To overcome this limitation, graphene oxide (GO) was incorporated into the ink formulated with crosslinkers to enhance reactivity. Porous polymers, such as CNF, have been employed for humidity sensing. To address the low reactivity caused by high resistance, we used Ag-dispersed Ag-CNFs. The fabricated temperature and humidity sensor exhibited a change of ΔR ≈ 0.14 with temperature variations between 20 °C and 100 °C at 20 RH% and a change of ΔC ≈ 2.8 with relative humidity variations between 20% and 80% in a 20 °C environment. These sensors can be designed for direct application on the surface of Li-ion batteries as they can be printed on flexible PET film and PI film, so they can be used in different EVs and IT devices, such as wearable devices.

## 2. Experiment

### 2.1. Preparation of Composite Materials for Humidity and Temperature Detection

CNF-Ag nanoparticles (NPs) were used as humidity-sensing materials. CNF-Ag NPs were synthesized from coarse CNF, AgNO_3_ (SAMCHUN CHEMICALS Co., Ltd., Seoul, Republic of Korea), and NaBH_4_ (Sigma–Aldrich, St. Louis, MO, USA). The synthesis process was performed as follows: coarse CNF and distilled water were mixed at a 1:2 weight ratio and then diluted through stirring for 1 h. Next, AgNO_3_ (0.02 mol/L) was added to the diluted solution and mixed for 4 h. Subsequently, a solution containing 1 mL of NaBH_4_ (0.1 M) and distilled water was added to the diluted solution and stirred for 24 h to produce CNF-Ag NP ink.

The temperature-sensing materials comprised PEDOT:PSS (Clevios P AI4083, Heraeus, Hanau, Germany), GOPS (Sigma–Aldrich, USA), and single-layer graphene oxide (Aqueous Solution, Graphene Supermarket, Ronkonkoma, NY, USA). A GOPS crosslinker was employed to enhance the moisture resistance of PEDOT:PSS [28]. To systematically investigate the humidity-sensing characteristics of GOPS-PEDOT:PSS, GOPS was added at different weight ratios. Assuming PEDOT:PSS to be 1, GOPS was added at ratios of 0, 0.75, and 1 and mixed by stirring for 30 min. GO was selected to improve the performance of the prepared GOPS-PEDOT:PSS temperature sensors. GO exhibits excellent thermal conductivity owing to its single-layer structure of carbon atoms arranged in a hexagonal lattice containing oxygen-containing functional groups [29,30,31,32,33]. For a performance comparison of the temperature sensor, assuming GOPS-PEDOT:PSS as 1, GO was added at ratios of 0, 0.3, 1, and 3 and mixed by stirring for 30 min. To minimize the impact of humidity, PDMS (SYLGARD^®^ 184, Sigma–Aldrich, USA) was chosen as the passivation layer for the fabricated GOPS-PEDOT:PSS + GO.

The structural characteristics of the fabricated CNF and Ag-CNF humidity sensors were analyzed using an X-ray diffractometer (XRD) and energy-dispersive X-ray analysis (EDAX). The structural properties were examined via XRD (D8 Discover, Bruker, Billerica, MA, USA) using monochromatic Cu-Kα radiation (λ = 1.5406 Å). The film thicknesses of the fabricated printed temperature–humidity sensors were determined using a scanning electron microscope (S-4830, HITACHI, Tokyo, Japan). Fourier-transform infrared spectroscopy (FTIR; iS50, Thermo Fisher Scientific, Waltham, MA, USA) and a high-resolution Raman/PL Spectrophotometer (LabRAM HR-800, HORIBA JOBIN YVON, Palaiseau, France) with a 785 nm excitation wavelength were used to determine the chemical composition and functional groups involved in the fabricated PEDOT:PSS and PEDOT:PSS with GOPS temperature sensors.

### 2.2. Fabrication of Integrated Temperature–Humidity Sensor

The proposed temperature and humidity sensors were manufactured using a 100% printing process. This printing process offers several advantages, including simplicity, low cost, and the ability to pattern various substrates. In addition, the minimal use of ink and solvents is environmentally friendly [34].

Figure 1a illustrates the fabrication process for the integrated temperature–humidity sensor. The substrate used was an external nylon film from an aluminum pouch, as previously described by Kim et al. [35,36]. Electrodes were created using an IDE structure and Ag ink (PG-007, PARU Co., Ltd., Sunchon, Republic of Korea) via screen printing. The gap, printing speed, and squeegee angle were 500 µm, 200 mm/s, and 39°, respectively. The printed electrodes were sintered at 150 °C for 30 min.

The fabricated IDE electrodes were coated with humidity- and temperature-sensing layers using Ag-CNF and PEDOT:PSS, respectively. This coating procedure was performed using a dispenser (Dispenser Shot Mini 200SX, MUSASHI, Tokyo, Japan) equipped with a nozzle (diameter: 0.19 mm and operated at a pressure of 50 kPa. The printed temperature–humidity sensing layers were then subjected to drying at 80 °C for 45 min. Subsequently, PDMS was applied to cover the humidity sensing layer via the dispenser, employing a pressure of 400 kPa, and then dried at 80 °C for 30 min. Figure 1b shows a visualization of the configuration of the fabricated temperature–humidity sensor.

### 2.3. Temperature–Humidity Sensors Measurement

The integrated temperature–humidity sensor was measured using an IMPEDANCE ANALYZER (IMPEDANCE ANALYZER 4192A, HP, Tokyo, Japan) and a temperature and humidity chamber (TEMP&HUMID CHAMBER, BSTECH Co., Ltd., Seoul, Republic of Korea), as shown in Figure 2. Temperature and humidity chambers were employed to measure both the resistance and capacitance characteristics in response to variations in the relative humidity and temperature. The relative humidity was incrementally increased in 20% increments, covering a range of 20–80%. Simultaneously, the temperature was elevated in 20 °C increments, spanning approximately 20 °C to 100 °C. The measurement frequency and Oscillator level were set at 1 kHz and 1 V, respectively.

## 3. Results and Discussion

### 3.1. Fabricated Temperature–Humidity Sensors

Figure 3a shows an IDE fabricated using the screen-printing process. The linewidth and spacing of the electrodes are measured as approximately 70 µm and 200 µm, respectively. The resistance of the IDE electrodes is varied within the range of approximately 0.3 to 0.8 Ω in length of 12 mm, as depicted in Figure 3b. Figure 3c shows the fabricated sensor and its cross-sectional scanning electron microscopy images. The electrode has an approximate thickness of 5.16 µm, while the temperature sensing layer, humidity sensing layer, and passivation layer have thicknesses of approximately 99.2 nm, 265 nm, and 10.4 µm, respectively.

### 3.2. Performance and Material Analysis of Humidity Sensors

Figure 4 shows the X-ray diffraction (XRD) spectra of the Ag-CNF and pristine CNF. Both Ag-CNF and pristine CNF exhibited a peak at approximately 22°, corresponding to the (002) plane [37,38,39,40]. Additionally, peaks for Ag were observed at 28°, 35°, 38.65°, 43°, and 46°, representing the (210), (122), (111), (200), and (231) planes, respectively [41,42,43,44]. Figure 5a,b shows the energy-dispersive X-ray analysis (EDAX) spectra of pristine CNF and Ag-CNF. The Ag-CNF contains approximately 21.15 wt% Ag.

Figure 6a,b shows the humidity-sensing characteristics of pristine CNF. As shown in Figure 6a, the pristine CNF responds at relative humidity levels above 60%. In Figure 6b, the response and recovery times are approximately 190 and 200 s, respectively, indicating slow reactivity. Conversely, Figure 6c shows the changes in resistance and capacitance of Ag-CNF in the relative humidity range of 20% to 80%. Changes in resistance and capacitance were observed in the lower humidity regions compared to pristine CNF. Figure 6d displays the recovery and response characteristics of Ag-CNF. The response times for resistance and capacitance were 32 and 26 s, respectively, and the recovery times were 155 and 114 s. Because water has a higher dielectric constant than air, the capacitance value increases with increasing humidity. Hence, the Ag-CNF film absorbs water, enhancing the mobility of hydrogen ions and reducing resistance [45,46]. Finally, Figure 6e,f shows the hysteresis for the resistance and capacitance of Ag-CNF. The changes in resistance and capacitance are approximately ΔR ≈ 0.06 and ΔC ≈ 2.7, respectively. These values indicate excellent responsiveness and low hysteresis, suggesting high reliability.

### 3.3. Performance and Material Analysis of Temperature Sensors

GOPS is a silane containing trimethoxysilane groups. This compound reacted with moisture to form a silica grid. GOPS molecules interacted with PEDOT:PSS, resulting in the formation of silane bonds. This interaction ultimately leads to a strong bond between the two substances, stabilizing PEDOT:PSS against moisture and enhancing its electrical properties [47].

Figure 7a shows the Fourier-transformed infrared (FTIR) spectra of the PEDOT:PSS and PEDOT:PSS + GOPS composites. When GOPS crosslinking agents were used to crosslink PEDOT:PSS, the FTIR analysis provided detailed information on the chemical interactions and functional groups within the resulting crosslinked materials. A broad peak appeared at 3100–3500 cm^−1^, related to the stretch of O–H, and another peak at approximately 1600–1700 cm^−1^ corresponds to the stretch of C=C present in both PEDOT:PSS and PEDOT:PSS + GOPS samples. Upon crosslinking with GOPS, several new peaks at approximately 905 cm^−1^, 1017 cm^−1^, 1095 cm^−1^, and 1450 cm^−1^ appear in the FTIR spectra of PEDOT:PSS + GOPS. The band appearing at approximately 1450 cm^−1^ is attributed to the stretching mode of C-C linked to thiophene rings, and that at 1095 cm^−1^ indicates the C-O-C bending vibration in the ethylenedioxy group. The bands at 1017 cm^−1^ and 905 cm^−1^ are assigned to the stretching of C-S. The creation of these new functional groups resulted from the crosslinking of GOPS in PEDOT:PSS [48]. The produced ink enables the acquisition of structural information through Raman spectroscopy (λ exc = 785 nm). Figure 7b shows that bands at 438, 988, and 1257 cm^−1^ are, respectively, assigned to C-O-C deformation, oxacyclic ring deformation, and C_α_–C_α_ inter-ring stretching. Particularly, the characteristic band at 1428 cm^−1^, attributed to symmetric C_α_ = C_β_ (-O) stretching, indicates a high degree of conjugation [49]. Furthermore, a shift from 1428 to 1438 cm^−1^ is observed, indicating a decrease in the concentration of charge carriers [50].

Figure 8a presents the moisture resistance of PEDOT:PSS at different concentrations of GOPS in the humidity range of 20% RH to 80% RH at 30 °C. When the ratio of PEDOT:PSS to GOPS was 1:1, moisture resistance decreased. Figure 8b illustrates the temperature sensor characteristics depending on the amount of added GOPS in the temperature range of 20 °C to 100 °C at 20% RH. In summary, no changes in the performance of the temperature sensor were observed even when the amount of GOPS added was increased. Therefore, a 1:1 ratio of GOPS to PEDOT:PSS was confirmed to maintain the performance of the temperature sensor while reducing moisture resistance.

Figure 9 presents the differential scanning calorimetry digital signal processing curve based on the weight ratio of GO added to PE-DOT:PSS-GOPS at a 1:1 ratio. When the ratio of PEDOT: PSS-GOPS to GO is 1:3, an endothermic peak is observed at temperatures ranging from approximately 30 to 40 °C. No endothermic peaks were detected at other ratios, indicating an improvement in thermal conductivity and enhanced characteristics of the temperature sensor. Table 1 lists the response and recovery times of the temperature sensor based on the PE-DOT: PSS-GOPS-to-GO to GO. When the GO ratio is 3, the response and recovery times are approximately 10.6 s and 10.4 s, respectively, exhibiting the best performance. Figure 10a–d illustrates the hysteresis of the temperature sensor as a function of GO ratio. The most effective GO ratio (3) was selected as the final temperature-sensing material. The results confirmed that this material has a low error in the hysteresis, indicating high reliability, and exhibits excellent reactivity with ΔR ≈ 0.17.

### 3.4. Performance of Integrated Temperature–Humidity Sensors

Figure 11a illustrates the characteristics of the integrated temperature–humidity sensor at 40 °C for humidity changes ranging from 20 to 80% RH. With the increase in humidity, significant changes of ΔC ≈ 2.8 and ΔR ≈ 0.08 are observed. Figure 11b represents the results when the humidity is fixed at 60% RH and the temperature varies from 20 to 100 °C. Changes of ΔC ≈ 1.17 and ΔR ≈ 0.14 are observed as the temperature increased. The thermal characteristics of the temperature-sensing materials were measured using a thermal imaging camera (C8940, FLIR, Stockholm, Sweden). Figure 12a shows the image of the fabricated sensor measured at 20 °C, and Figure 12b shows the image measured at 80 °C. These results confirm the excellent thermal characteristics of the temperature-sensing section owing to the addition of GO.

## 4. Conclusions

In this study, an integrated temperature–humidity sensor was fabricated on the pouch film of a pouch-type battery using a 100% printing process. The sensor made with the Ag-CNF thin film exhibited superior reactivity to changes in humidity compared with traditional CNF. The temperature sensor fabricated from a PEDOT:PSS-based thin film was integrated into a single IDE. Measurements were conducted in temperature- and humidity-controlled chambers. The fabricated integrated sensor demonstrates excellent reactivity in a temperature range of 20 to 100 °C and a humidity range of 20 to 80% RH. These results offer a new methodology for simultaneous temperature and humidity measurements and are expected to be applicable to different Li-ion battery monitoring applications to preemptively address the risk of battery explosions.

## Figures and Tables

**Figure 1 sensors-24-00104-f001:**
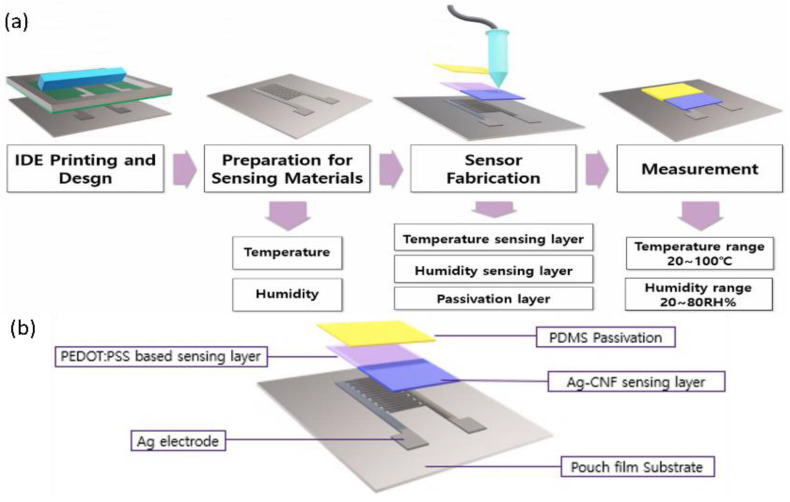
(**a**) Fabrication process of printed temperature–humidity sensor, and (**b**) schematic of the printed temperature–humidity sensor structure.

**Figure 2 sensors-24-00104-f002:**
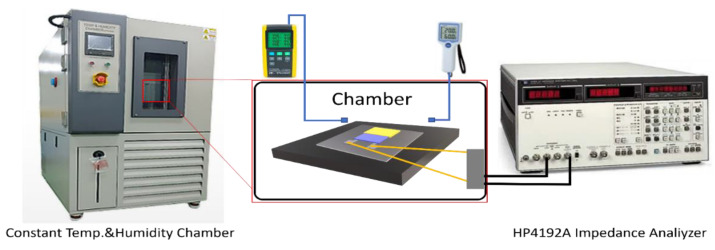
Schematic for sensor measurements.

**Figure 3 sensors-24-00104-f003:**
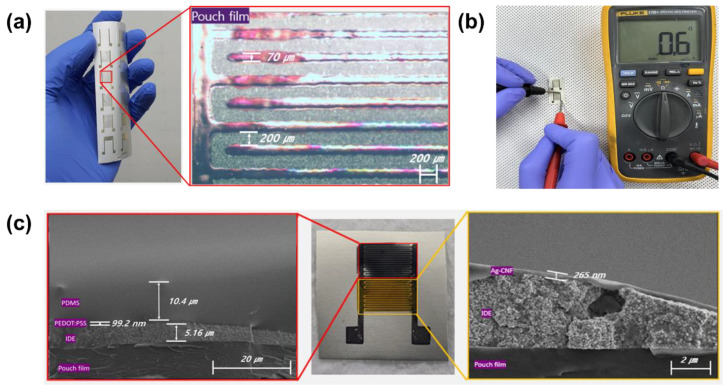
(**a**) Photograph and optical microscope (OM) image of printed IDE on pouch film. (**b**) Image of resistance measurement of the fabricated electrode. (**c**) Cross-sectional scanning electron microscopy (SEM) images of printed temperature–humidity sensor.

**Figure 4 sensors-24-00104-f004:**
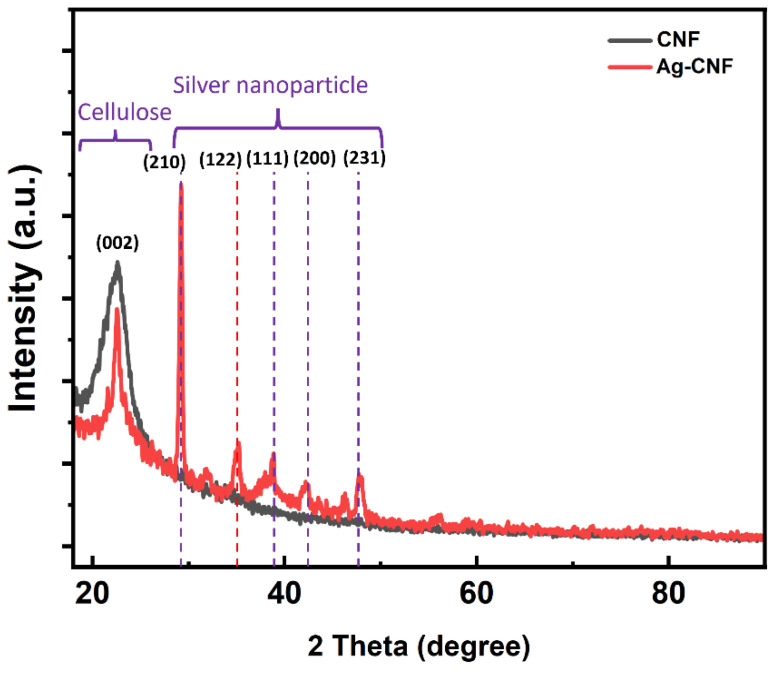
XRD patterns of CNF and Ag-CNF.

**Figure 5 sensors-24-00104-f005:**
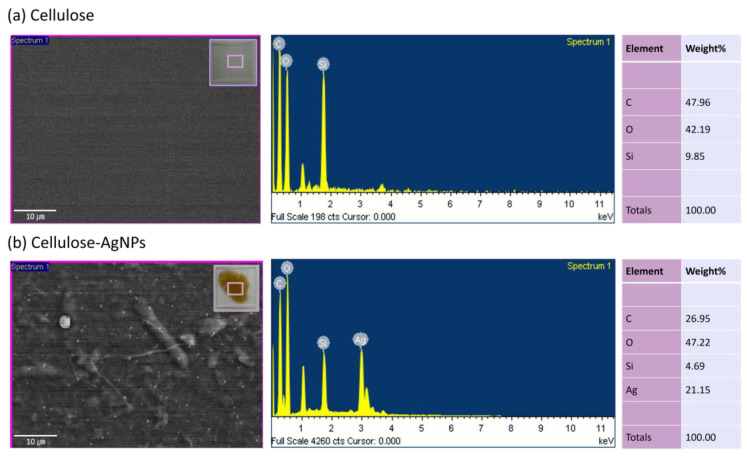
EDAX spectrum (**a**) of cellulose, and (**b**) cellulose-Ag-NPs.

**Figure 6 sensors-24-00104-f006:**
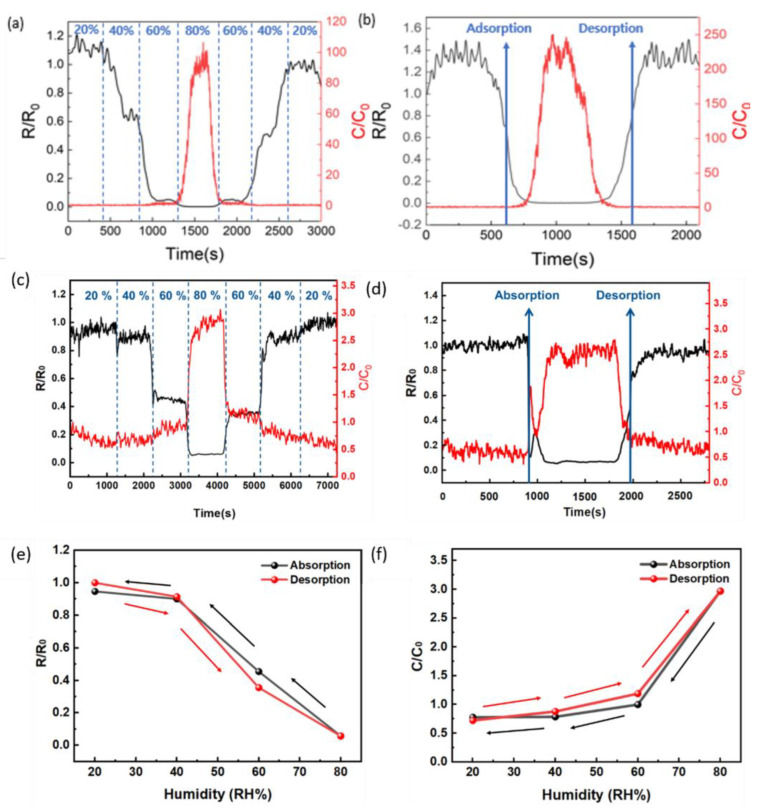
(**a**) Electrical response of the CNF humidity sensor based on measurements at a temperature of 30 °C, with the humidity change sequentially varying from 20 RH% to 80 RH%. (**b**) Transient electrical responses of the printed CNF thin film sensor in the humidification/dehumidification process. The electrical response of the sensor as a function of humidity varies from 20 RH% to 80 RH%. (**c**) The electrical response of the Ag-CNF humidity sensor. (**d**) Transient electrical responses of the printed Ag-CNF thin film sensor. The electrical response of the sensor as a function of humidity varies from 20 RH% to 80 RH%. (**e**) Resistance reactivity. (**f**) Capacitive reactivity.

**Figure 7 sensors-24-00104-f007:**
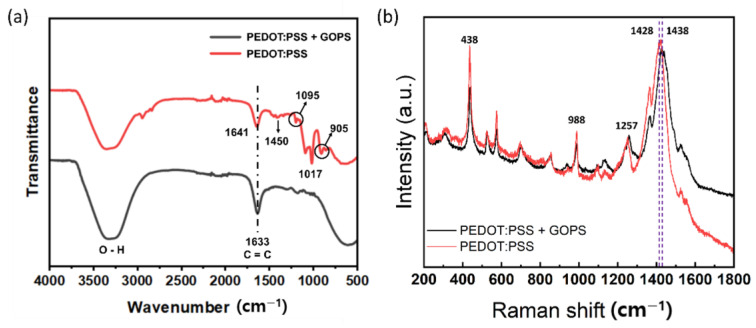
(**a**) FTIR spectra of PEDOT:PSS and PEDOT:PSS with GOPS, and (**b**) Raman spectra of PEDOR:PSS, PEDOT:PSS + GOPS.

**Figure 8 sensors-24-00104-f008:**
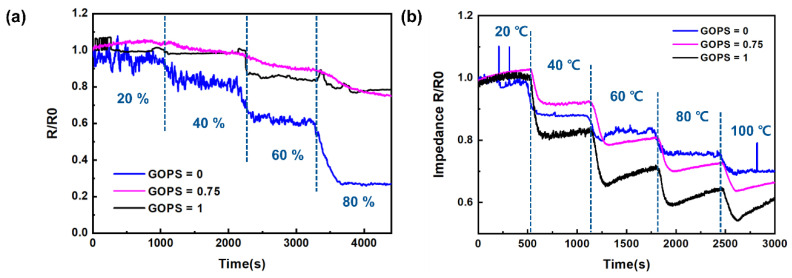
(**a**) Humidity-dependent relative resistance changes of PEDOT:PSS films with different concentrations of GOPS (0, 0.75, and 1). (**b**) Temperature-dependent relative resistance changes of PEDOT:PSS films with different concentrations of GOPS (0, 0.75, and 1). Inks with GOPS to PEDOT:PSS weight ratios of 0, 0.75, and 1 are denoted as GOPS = 0, GOPS = 0.75, and GOPS = 1, respectively.

**Figure 9 sensors-24-00104-f009:**
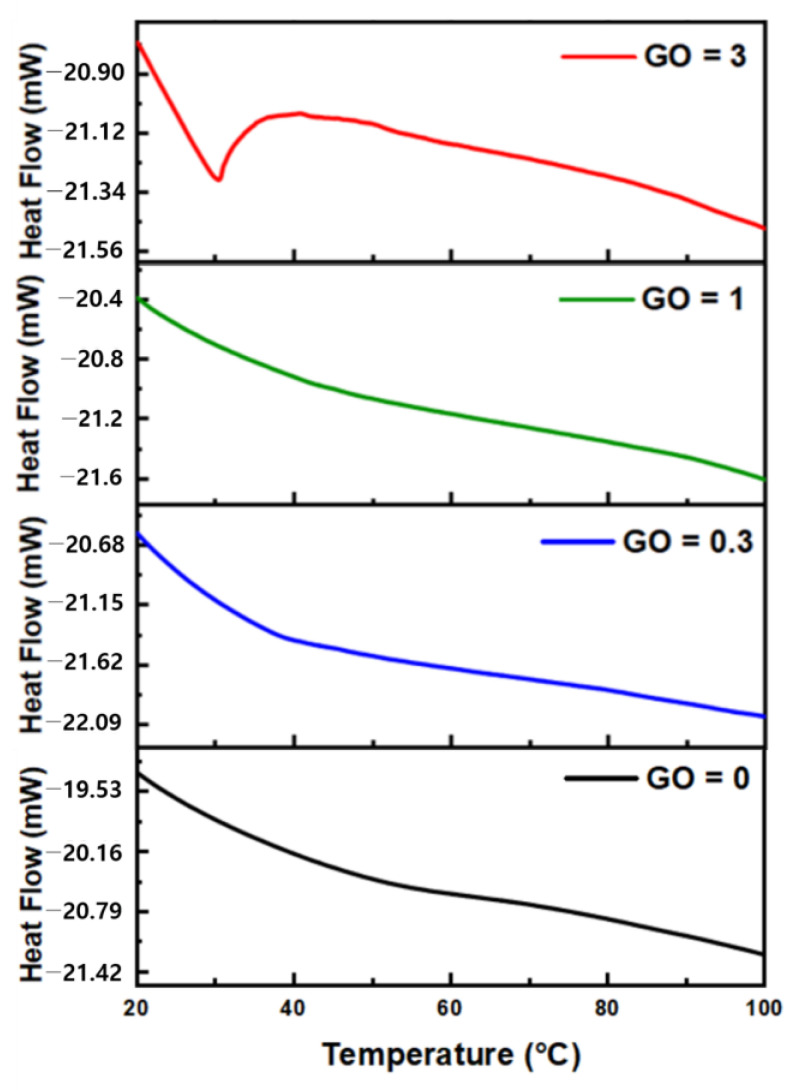
Heat flow of different ratios of GO: 0, 0.3, 1, and 3. PEDOT:PSS + GOPS to GO weight ratio of 0, 0.3, 1, and 3 are denoted as GO = 0, GO = 0.3, GO = 1, and GO = 3, respectively.

**Figure 10 sensors-24-00104-f010:**
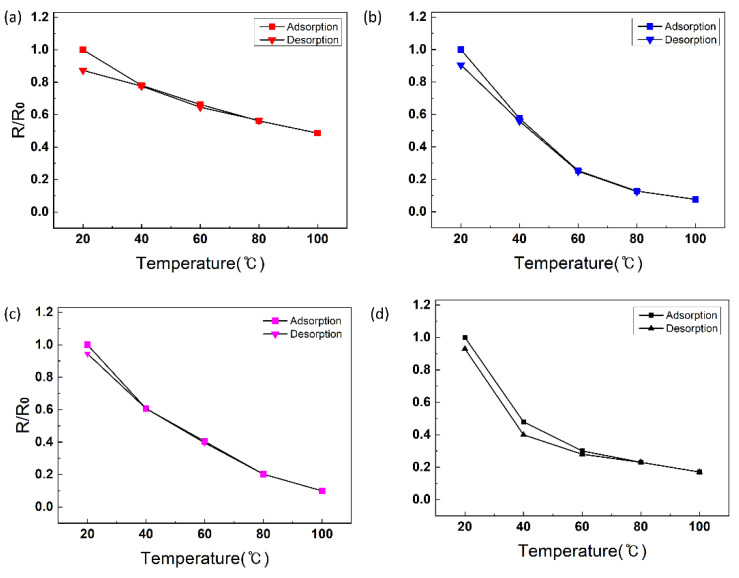
Hysteresis of selected temperature sensing materials. Changes in the relative resistance of the temperature sensor according to the addition weight ratio of GO (**a**) 0, (**b**) 0.3, (**c**) 1, and (**d**) 3.

**Figure 11 sensors-24-00104-f011:**
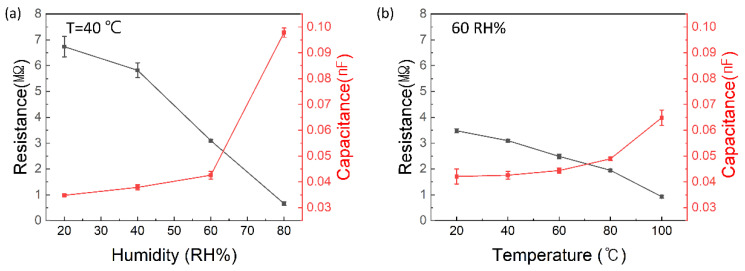
Electrical response of the humidity temperature sensors according to the measurement results obtained by varying the humidity within the range of 20%, 40%, 60%, and 80% RH and temperature range of 20 to 100 °C. Performance of (**a**) variation in electrical resistance and capacitance of the temperature–humidity sensor with humidity at 40 C, and (**b**) variation in electrical resistance and capacitance of the sensor with temperature at RH = 60%.

**Figure 12 sensors-24-00104-f012:**
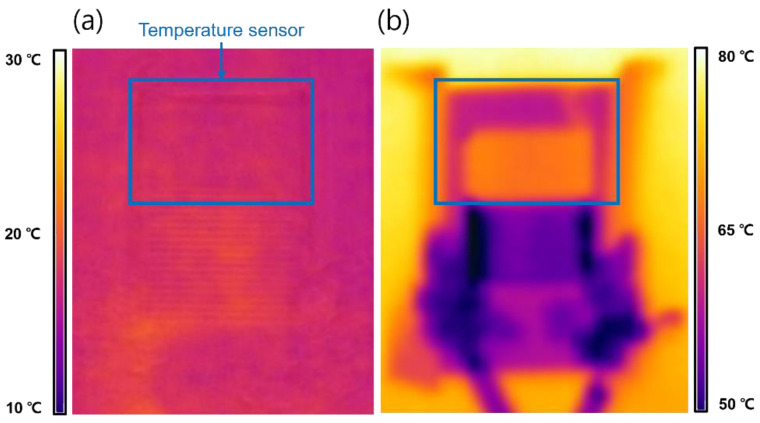
(**a**) Thermal image of fabricated sensors at 20 °C. (**b**) Thermal image of fabricated sensors at 80 °C.

**Table 1 sensors-24-00104-t001:** Changes in the response and recovery times with different weight ratios of GO.

	GO = 0	GO = 0.3	GO = 1	GO = 3
Response time (s)	13.6 s	12.8 s	12.4 s	10.6 s
Recovery time (s)	15.6 s	12.6 s	11.6 s	10.4 s

## Data Availability

Data are contained within the article.

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
