# Peer review of "Integrated Temperature–Humidity Sensors for a Pouch-Type Battery Using 100% Printing Process"

_sensors, 2023, doi:10.3390/s24010104_

Round 1
Reviewer 1 Report
Comments and Suggestions for Authors
This paper reported integrated temperature-humidity sensors, which can be potential for efficient temperature and humidity monitoring applications in various pouch-type lithium-ion batteries. Some issues may be improved for publication.
1. There are too many paragraphs in the Introduction section, and it is recommended to merge them appropriately.
2. The test resistance in Figure 3 is recommended to remain uniform, because the length and width of the device also affect the internal resistance.
3. The Raman measurement of PEDOT:PSS and PEDOT:PSS with GOPS may be provided.
4. The author introduces a lot of lithium-ion batteries in the body part, but in the end the sensor does not use this above.
Comments on the Quality of English LanguageSome minor revision may be necessary.
Author Response
All the points raised by the reviewer are taken into consideration, and accordingly, the manuscript is amended. The authors would like to thank the reviewers for improving the quality of the present manuscript. All the amendments/additions made in the revised manuscript are highlighted in Yellow.
Reviewer 1
- There are too many paragraphs in the Introduction section, and it is recommended to merge them appropriately.
Answer 1. As per the suggestion of the reviewer, the paragraphs in the introduction section are merged in the revised manuscript.
- The test resistance in Figure 3 is recommended to remain uniform, because the length and width of the device also affect the internal resistance.
Answer 2. The test resistance is remained same in whole the measurement and the length of the IDE is incorporated in the revised manuscript.
- The Raman measurement of PEDOT:PSS and PEDOT:PSS with GOPS may be provided.
Answer 3. The Raman spectra of PEDOT:PSS and PEDOT:PSS with GOPS are measured and the necessary explanations are incorporated in the revised manuscript.
- The author introduces a lot of lithium-ion batteries in the body part, but in the end the sensor does not use this above.
Answer 4. As per the suggestion of reviewer, the frequent use of the word “lithium-ion batteries ” is avoided in the current revised manuscript. In the current study, we have designed the temperature sensor using graphene oxide (GO) and PEDOT: PSS composites and humidity sensor using Ag-CNFs on the surface of a pouch film using a fully printing process and studied its sensitivity.
This fabricated sensor can be used in the direct application on the surface of Li-ion batteries. We consider as it our future work and our goal is to apply it to real batteries and ensure that the sensors applicability.

Reviewer 2 Report
Comments and Suggestions for Authors
The presented research work is interesting, and it is within the scope of this journal.
1) the introduction should be improved with more recent research work in this domain. And please point out the main challenges and the main contributions of this paper in this domain.
2) The proposed sensor seems to have interesting application for battery diagnosis and prognosis. Is it possible to present some research link or perspective with some control-oriented research work, such as A degradation empirical-model-free battery end-of-life prediction framework based on gaussian process regression and Kalman filter.
3) As for the application level, the proposed method has some limitations ? For example, difficult to extend to large production requirement ? (or are there some other critical limitations that can prevent the development of your proposed method ? )
Author Response
All the points raised by the reviewer are taken into consideration, and accordingly, the manuscript is amended. The authors would like to thank the reviewers for improving the quality of the present manuscript. All the amendments/additions made in the revised manuscript are highlighted in Yellow.
Reviewer 2
1) the introduction should be improved with more recent research work in this domain. And please point out the main challenges and the main contributions of this paper in this domain.
Answer 1. As per the suggestion of the reviewer, the introduction is improved with some addition of recent research work and necessary citation is included in the revised manuscript.
2) The proposed sensor seems to have interesting application for battery diagnosis and prognosis. Is it possible to present some research link or perspective with some control-oriented research work, such as A degradation empirical-model-free battery end-of-life prediction framework based on gaussian process regression and Kalman filter.
Answer 2. As per the suggestion of the reviewer, some control-oriented research work is included n the revised manuscript.
3) As for the application level, the proposed method has some limitations ? For example, difficult to extend to large production requirement ? (or are there some other critical limitations that can prevent the development of your proposed method ? )
Answer 3. In this study, printing was conducted directly on pouch film intended for batteries; however, it is feasible to employ a method of printing on various films and subsequently attaching them. The manufacturing process for pouch film involves a printing process utilizing dry lamination, thereby facilitating mass production when applied to the pouch film manufacturing process.
